# WaveDN: A Wavelet-based Training-free Zero-shot Enhancement for Vision-Language Models

## ABSTRACT

Vision-Language Models (VLMs) built on contrastive learning, such as CLIP, demonstrate great transferability and excel in downstream tasks like zero-shot classification and retrieval. To further enhance the performance of VLMs, existing methods have introduced additional parameter modules or fine-tuned VLMs on downstream datasets. However, these methods often fall short in scenarios where labeled data for downstream tasks is either unavailable or insufficient for fine-tuning, and the training of additional parameter modules may considerably impair the existing transferability of VLMs. To alleviate this issue, we introduce WaveDN, a wavelet-based distribution normalization method that can boost the VLMs' performance on downstream tasks without parametric modules or labeled data. Initially, wavelet distributions are extracted from the embeddings of the sampled, unlabeled test samples. Subsequently, WaveDN conducts a hierarchical normalization across the wavelet coefficients of all embeddings, thereby incorporating the distributional characteristics of the test data. Finally, the normalized embeddings are reconstructed via inverse wavelet transformation, facilitating the computation of similarity metrics between the samples. Through extensive experiments on two downstream tasks, using a total of 14 datasets covering text-image and text-audio modal data, WaveDN has demonstrated superiority compared to state-of-the-art methods.

## CCS CONCEPTS

• **Computing methodologies → Computer vision**.

## KEYWORDS

vision language, test-time augmentation, multi-modal, zero-shot recognition, wavelet transform

## 1 INTRODUCTION

Large-scale self-supervised pre-training has been extensively validated as highly effective and reliable through notable contributions such as GPT-4[1], ERNIE[2], Sora[3], and CLIP[4]. The use of pre-trained models on open-set tasks has shown promising results, with transferability that rivals supervised methods designed for closed-set tasks. In the field of computer vision, Contrastive Language-Image Pre-training (CLIP)[4] has demonstrated its efficient and accurate open-set recognition capability across various

*ACM MM, 2024, Melbourne, Australia*

© 2024 Copyright held by the owner/author(s). Publication rights licensed to ACM.
ACM ISBN 978-x-xxxx-xxxx-x/YY/MM
https://doi.org/10.1145/nnnnnnn.nnnnnnn

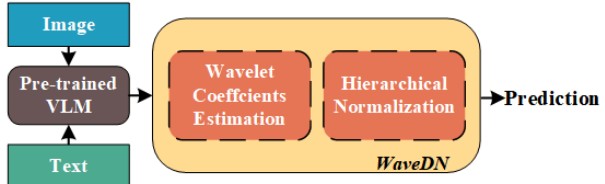

**Figure 1: Training-free zero-shot enhancement using WaveDN . We propose *WaveDN*, a method for enhancing the zero-shot transfer abilities of VLMs, without training. By performing wavelet transformation and coefficient feature extraction on *a small amount of unlabeled test data embeddings*, the estimated wavelet distribution coefficients are obtained. Before calculating similarity, utilize the estimation coefficients from the previous stage to transform each embedding using hierarchical normalization, thereby enhancing performance during testing.**

visual tasks such as image classification, instance segmentation, object detection, and more downstream tasks. CLIP is trained on an extensive image-text corpus utilizing the InfoNCE Loss[5, 6], which enhances the similarity among positive sample pairs while diminishing the similarity among negative sample pairs, thereby establishing a meaningful alignment between the visual and linguistic modalities. During model evaluation, solely the dot product between image and text features is requisite for computing their similarity.

Presently, a multitude of advancements have been implemented to refine CLIP, including methods rooted in few-shot learning[7–9] and model fine-tuning[10–12], all aimed at augmenting CLIP's efficacy across downstream tasks. Nevertheless, these strategies, which often necessitate supplementary parameter modules or rely on annotated test data, pose significant obstacles to the inherent efficiency and generalization capacity of CLIP. To circumvent the introduction of extra parameters and labeled data, testing enhancement techniques like CALIP[13] and Distribution Normalization(DN)[14] have been introduced. These methods focus on enhancing the zero-shot capability of the model by refining and optimizing the features extracted by the text and image encoders.

The research of DN[14] indicates that, during training, the model takes into account information from both positive and negative sample pairs. However, during testing, using only the dot product between two modals' embeddings is a zeroth-order approximation of the InfoNCE loss. The absence of distribution data from the test dataset in similarity calculations results in information loss. This dot product-based similarity calculation method has not fully utilized the model's capabilities. The DN method introduces the mean of test samples during the similarity calculation phase to align the loss functions between testing and training.

However, while this method partially addresses the issue of misaligned testing loss functions, performing operations on features with high semantic information, such as subtracting the mean, will lead to additional information loss. The features of the test sample will be compromised due to additional processing. The results in Figure. 2 can better illustrate this point. After implementing the DN, the proportion of classes exhibiting positive effects surpasses those with negative effects, yielding an enhancement in overall recognition accuracy at the dataset level. Nonetheless, at the individual class accuracy level, a considerable fraction of categories exhibit reduced recognition accuracy, with reductions reaching up to 30%. This decline is attributed to substantial degradation in the representation quality of the test data.

To tackle this issue, we introduce WaveDN, which employs wavelet transformation to decompose the test samples' embeddings, facilitating a more detailed hierarchical normalization process that maximally preserves the features of the test samples when introducing the test data distribution. The method's brief process is shown in Figure.1. Initially, a wavelet decomposition based wavelet coefficient mean estimation is performed. Similar to DN, WaveDN's mean estimation only requires sampling a small portion of unlabeled test data to obtain a convergent estimation result, as confirmed by experiments. Subsequently, during the inference stage, the previous stage's wavelet coefficients are used to hierarchically normalize the embeddings of the text or images to be predicted. In this process, each embedding is transformed into multiple wavelet coefficients, providing a more detailed reflection of its information. A weighted normalization using wavelet coefficient similarity is then conducted, aiming to preserve the information content of the test data while introducing data distribution information. As shown in Figure.2, WaveDN has a lesser negative impact on fewer classes and a positive impact on more classes, thereby to some extent avoiding representation degradation of test samples and achieving better utilization of test data.

The overall process of the WaveDN framework is zero-shot, parameter-free, and does not require additional training, which is consistent with CALIP[13] and DN[14]. Wavelet transformation can effectively handle encoding results with high information density. By considering embedding generated by vision-language models as a time series signal and performing wavelet decomposition, information at different frequencies and times can be obtained, providing a more precise and reliable reference for subsequent transformations. Through our meticulous calculations, WaveDN has demonstrated the capability to strike a delicate equilibrium between aligning the loss function through the incorporation of negative sample information and mitigating information loss in normalized test samples, consequently mitigating the adverse effects of sample mean introduction. As depicted in Figure 2, our methodology demonstrates enhanced efficacy relative to DN, achieving favorable results devoid of significant adverse effects, such as a marked reduction in recognition accuracy across specific categories. WaveDN not only augments the overall accuracy of category recognition but also preserves the generalization capabilities of VLMs without undermining their robust generalization capacity.

To substantiate the efficacy and supremacy of our approach, we employed WaveDN in the context of two downstream tasks: Zero-shot classification and Cross-modal Retrieval, juxtaposed against

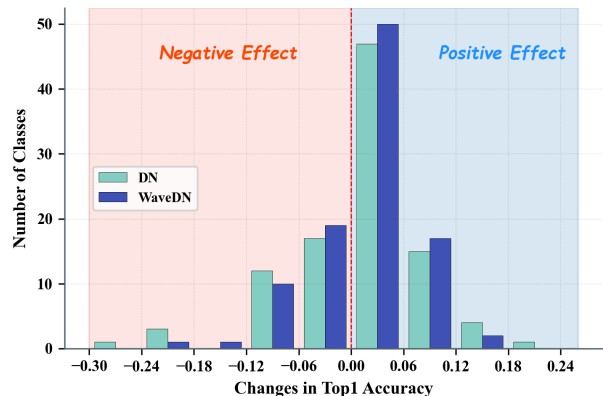

Figure 2: Impact of Top1 Accuracy on different classes of Cifar100. The height of each column represents the number of categories influenced to some extent. The horizontal axis values represent the impact on the top 1 accuracy of the class level after using a certain method. Negative values indicate a decrease in class accuracy after using the method, while positive values indicate improvement.

cutting-edge methodologies. Our experimental framework encompassed the utilization of diverse CLIP visual encoder architectures, with a focus on augmenting modal alignment across text, image, and audio pre-training encoders. Findings derived from evaluations conducted on 12 image datasets and 2 audio datasets unequivocally underscore the overarching superiority of WaveDN. Notably, for some benchmarks, it surpassed certain existing approaches that necessitate few-shot fine-tuning. Furthermore, comprehensive ablation analyses corroborate the computational efficiency and the relatively straightforward and robust implementability of our methodology.

The main contributions of WaveDN:

- WaveDN creatively employs wavelet transformation to decompose the information-rich embeddings extracted by the VLM model, supporting more detailed analysis and computations for distribution normalization.
- Wavelet-based feature extraction and hierarchical normalization can better introduce the test data distribution during the testing phase. This approach aligns with the InfoNCE loss for VLM, while maintaining the fidelity of the original data.
- Experiments across a multitude of datasets has substantiated the superior performance and multimodal scalability of the WaveDN. WaveDN is a parameter-free, training-free, and computationally efficient VLM zero-shot enhancement method.

## 2 RELATED WORK

### 2.1 Vision-Language models' downstream augmentation

This zero-shot transfer capability positions CLIP[4] on par with supervised models tailored for specific tasks. Efforts to augment the

prowess of Visual Language Models have spurred the development of innovative methodologies. Prompt-based approaches[7, 9, 15], for instance, strive to refine input data, bridging the gap between the distribution of test data and that of the training data for VLMs. Fine-tuning techniques[10, 11, 16] have also emerged as a means to enhance performance on downstream tasks while preserving the original generalization capacities of VLMs. In a bid to enhance the accessibility and versatility of models, researchers have proposed parameter-free and training-free methods. Noteworthy examples include CALIP[13], which introduces a parameter-free attention mechanism, and SuS-X[17], which adopts a name-only strategy to construct image support sets. There are also some works[18, 19] that leverage the power of large language models to enrich text labels, maximizing information utilization during testing.

Moreover, Distribution normalization[14] introduces a novel approach aligning InfoNCE loss[20] during testing, effectively harnessing distribution information from test data without the need for labeled data. In this paper, we explore a distribution estimation approach utilizing wavelet decomposition, accompanied by a hierarchical normalization method that exerts minimal impact on the features of test data relative to the DN method.

## 2.2 Wavelet method for Deep learning

Currently, many studies utilize Discrete Wavelet Transform (DWT) to enhance the efficiency of deep learning models. The Grid-based neural radiation field wavelet transformation method [21] diminishes the computational demands of neural rendering while preserving the benefits of supplementary data structures. Discriminative Wavelet Sub-bands [22] utilize unsampled 2D discrete wavelet transformation for the recognition of deformed facial images. HifaFace [23] employs wavelet transformation to convert images into multiple frequency domains, wherein the high-frequency components facilitate the recovery of detailed features. NEGAN, a noise prior learner [24], predominantly aligns the high-frequency segments of noisy images using discrete wavelet transformation (DWT). DaReNeRF [25] implements Inverse Dual-Tree Complex Wavelet Transformation (DTCWT) to reconstruct plane-based information, with features for each spatiotemporal point derived by amalgamating vectors from these reconstituted planes. The DWT-FFC frequency branch [26] leverages DWT to extract complex high-frequency features. In our research, we treat the embeddings produced by VLM encoders as one-dimensional discrete time series signals. We apply discrete wavelet transformation to these signals, extracting the wavelet distribution coefficients. This process decomposes embeddings, which are rich in semantic information, into finer scales, thereby facilitating more precise calculations and hierarchical normalization operations.

## 3 METHOD

### 3.1 Preliminary

*3.1.1 CLIP.* CLIP is a large-scale visual language model pre-trained on 400 million text-image pairs. To achieve cross-modal feature extraction and prediction, CLIP consists of a text encoder and a visual encoder, donated as $E_T$ and $E_I$. The text segment employs a transformer encoder[27] for text encoding, whereas the image segment utilizes ResNet[28] or Vision-transformer[29] structures

for image encoding, enabling it to excel in cross-domain image and text processing. For an image $i$ and a text $t$, the image embedding $I \in R^{1 \times C}$ and text embedding $T \in R^{1 \times C}$ can be obtained as follows:

$$I = E_I(i) \tag{1}$$

$$T = E_T(t) \tag{2}$$

As shown in Eq. 3, the similarity between text embedding and image embedding is calculated through dot product. In downstream tasks, the similarity used for prediction is obtained through the dot product of text embeddings and image embeddings after L2 normalization.

$$M_{org} = L2(I)^T \cdot L2(T) \tag{3}$$

For the image recognition task, the goal is to determine which of the $K$ categories the test image belongs to. The text prompt contains the names of the $K$ categories as tokens. The encoded $F_T = \{T_1, T_2, \ldots, T_K\}$ contains $K$ embeddings, where the j-th vector in the set $F_T \in R^{K \times C}$ represents the features of prompts containing the name of the j-th category. For a test image embedding $I_1 \in R^{1 \times C}$. Calculating similarity using Eq.3 results in a similarity matrix $M \in R^{1 \times C}$, where the text prompt corresponding to the highest similarity value in the matrix is the classification result for this image.

*3.1.2 Distribution Normalization.* During training, CLIP utilizes the Information Noise Contrastive Estimation(InfoNCE)[5, 6] loss function, which considers both positive and negative sample pairs. However, during testing, only the dot product calculation using Eq. 3 is used as a similarity calculation method, which has been proven in DN[14] to be a zeroth-order approximation of InfoNCE loss. This research also highlights that using zeroth-order approximation to calculate similarity leads to the loss of information regarding the distribution of negative samples, which is crucial for predictions. To alleviate this issue, during the prediction phase, the DN method replaces the approach of Eq. 3 with the similarity calculation method from Eq. 4 to introduce distribution information of negative sample pairs, which has been proven to be a reliable first-order approximation of InfoNCE loss. $\mu_I$ and $\mu_T$ are estimated mean values obtained through sampling a small number of test samples' embeddings.

$$M_{DN} = L2(I - \frac{1}{2}\mu_I)^T \cdot L2(T - \frac{1}{2}\mu_T) \tag{4}$$

*3.1.3 Wavelet transform.* Wavelet decomposition provides a powerful tool for processing and analyzing various signals by offering a method for local analysis in both time and frequency domains simultaneously. Its multi-scale nature demonstrates advantages that traditional methods cannot match when dealing with non-stationary signals with complex frequency characteristics. The ultimate goal of wavelet decomposition is to decompose any time-domain signal $x(t)$ into multiple wavelet coefficients containing different information of the signal, including approximation coefficients $\{A_l, A_{l-1}, \ldots, A_1\}$, and detail coefficients $\{D_l, D_{l-1}, \ldots, D_1\}$ where $l$ represents the number of wavelet decomposition levels. Once the wavelet basis is selected, the low-pass filter coefficients $h_k$ and high-pass filter coefficients $g_k$ can be determined. The first-level decomposition in the iterative process of wavelet decomposition is as follows:

$$A_1 = \sum_k h_k * x(2t - k) \tag{5}$$

$$D_1 = \sum_k g_k * x(2t - k) \tag{6}$$

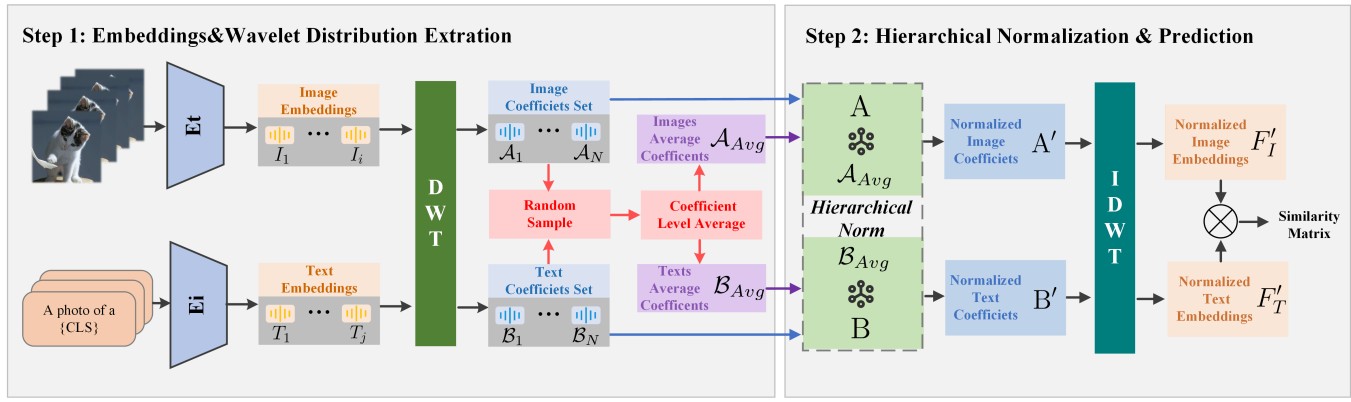

**Figure 3: The pipeline of WaveDN. We introduce WaveDN, a parameter-free zero-shot enhancement method for VLM. During testing, we apply the *Discrete Wavelet Transformation(DWT)* to the embeddings of multimodal data to obtain the wavelet coefficient set, then sample a small number of coefficients to obtain the mean wavelet distribution of the test data. Then, normalize the wavelet coefficients of all data using a *hierarchical normalization algorithm*. Finally, WaveDN restores the data representation to its original dimensions through *Inverse Discrete Wavelet Transformation(IDWT)* for similarity calculation.**

For each subsequent level $u(2 \leq u \leq l)$, the same decomposition process is applied to the previous approximation coefficient $cA_{u-1}$ to obtain new approximation coefficient $cA_u$ and detail coefficient $cD_u$:

$$A_u = \sum_k h_k * A_{u-1}(2t - k) \quad (7)$$

$$D_u = \sum_k g_k * D_{u-1}(2t - k) \quad (8)$$

The highest level wavelet approximation coefficient $A_l$ and all wavelet detail coefficients $\{D_l, D_{l-1}, \ldots, D_1\}$ can represent all the information of the original signal. In the following text, we can represent the process of discrete wavelet transformation(DWT) as Eq. 9, where the embedding $v$ output by the encoder is decomposed through DWT into $l + 1$ wavelet coefficients $\mathcal{E} = \{e_1, e_2, \ldots, e_{l+1}\}$.

$$\mathcal{E} = \{A_l, D_l, D_{l-1}, \ldots, D_1\} = DWT(v) \quad (9)$$

The inverse discrete wavelet transform(IDWT) can be performed by combining the scaling function $\theta_{j,k}(t)$ and wavelet function $\varphi_{j,k}(t)$ of the selected wavelet basis as shown in Eq. 10. In the following text, we can abbreviate this process as shown in the form of Eq. 11

$$x(t) = \sum_k A_{l,k} * \theta_{l,k}(t) + \sum_{j=1}^{l} \sum_k D_{j,k} * \varphi_{j,k}(t) \quad (10)$$

$$v = IDWT(\mathcal{E}) \quad (11)$$

## 3.2 Frameworks of WaveDN

Introducing a distribution with negative sample information is crucial during the testing phase to align with InfoNCE loss. However, it is essential to minimize the disruption of the rich information content in the embeddings. Based on this concept, we propose the WaveDN method, as illustrated in Figure. 3. It involves two main steps: extracting wavelet distributions and hierarchical normalization. The former aims to estimate a distribution with less information loss compared to directly taking the mean. The latter

incorporates this distribution estimation into the similarity calculation process during the testing phase to utilize information from the negative samples while also ensuring the maximum preservation of the test sample' embeddings.

*3.2.1 Wavelet Distribution Extraction.* By sampling a finite number of unlabeled samples, we can obtain vectors that approximate convergence and contain distribution characteristics. The experimentation regarding the sampling quantity is discussed in the section 4.4.1. After random sampling, $N$ images and text are obtained, which are then processed through the corresponding encoder for feature extraction, resulting in an image embedding set $\{I_1, I_2, \ldots, I_N\}$ and a text embedding set $\{T_1, T_2, \ldots, T_N\}$. By performing DWT with $l$ level to an embedding $I_i$ in the image feature sets, it can be decomposed into a set with $l + 1$ wavelet coefficients, donated as $\mathcal{A}_i = \{\alpha_{i,1}, \alpha_{i,2}, \ldots, \alpha_{i,l+1}\}$. The text feature set undergoes the same transformation as described above. A text feature $T_i$ will be decomposed as $\mathcal{B}_i$. The process can be represented as:

$$\mathcal{A}_i = DWT(I_i) = \{\alpha_{i,1}, \alpha_{i,2}, \ldots, \alpha_{i,l+1}\} \quad (12)$$

$$\mathcal{B}_i = DWT(T_i) = \{\beta_{i,1}, \beta_{i,2}, \ldots, \beta_{i,l+1}\} \quad (13)$$

Decomposing the embedding into multiple wavelet coefficients, each representing partial information of the embedding at different scales and positions. Through wavelet decomposition, the embedding with high information density is to some extent unfolded. The process of extracting the mean distribution is to take the average of wavelet coefficients at the same level. By performing operations as shown in Eq. 14 and Eq. 15, wavelet average distributions of images $\mathcal{A}_{Avg}$ and texts $\mathcal{B}_{Avg}$ are obtained for the hierarchical normalization in the next step. This feature extraction method, first performing DWT and then taking the mean, can maximize the preservation of local information for each sample. Compared to directly averaging the embedding, which can only provide overall average information, our method can achieve more accurate distribution feature extraction.

$$\mathcal{A}_{Avg} = \{ \frac{\sum_{i=1}^{N} \alpha_{i,1}}{N}, \frac{\sum_{i=1}^{N} \alpha_{i,2}}{N}, \dots, \frac{\sum_{i=1}^{N} \alpha_{i,l+1}}{N} \} \quad (14)$$

$$\mathcal{B}_{Avg} = \{ \frac{\sum_{i=1}^{N} \beta_{i,1}}{N}, \frac{\sum_{i=1}^{N} \beta_{i,2}}{N}, \dots, \frac{\sum_{i=1}^{N} \beta_{i,l+1}}{N} \} \quad (15)$$

*3.2.2 Hierarchical normalization.* The main process of hierarchical normalization is shown in Figure. 3 on the right. For a general task of downstream inference, the similarity matrix between $i$ image embeddings $F_I = \{I_1, I_2, \dots, I_i\}$ and $j$ text embeddings $F_T = \{T_1, T_2, \dots, T_j\}$ needs to be calculated. Initially, transform all embeddings into coefficients sets using DWT, resulting in image sets A and text sets B.

$$A = \{\mathcal{A}_1, \dots, \mathcal{A}_i\} = \{DWT(I_1), \dots, DWT(I_i)\} \quad (16)$$

$$B = \{\mathcal{B}_1, \dots, \mathcal{B}_j\} = \{DWT(T_1), \dots, DWT(T_j)\} \quad (17)$$

After performing hierarchical normalization computation as shown in Algorithm 1, the normalized coefficient sets A′ and B′ are obtained. Specifically, the hierarchical normalization operation involves applying a weighted normalization to each wavelet coefficient sequence in the wavelet coefficient set, calculating the similarity between the wavelet coefficients of each layer of the test sample and the mean wavelet coefficients extracted earlier for the corresponding layer as the normalization weight. Also, there is a fixed constant $\lambda$ that will be used in the normalization process, whose value is determined through experiments.

By performing weighted normalization of wavelet coefficients at different levels, our method can execute normalization operations of different intensities at different levels. For a certain level of wavelet coefficients, if its similarity with the mean wavelet coefficients of the corresponding level is lower, it indicates that this level contains more information representing the sample itself, which should be preserved to a greater extent. Through this hierarchical normalization with different intensities, our method can introduce distribution information while retaining more specific information about the test samples.

Using IDWT, transform the variables from $\mathcal{A}'_k$ and $\mathcal{B}'_k$ obtained above to the same dimensions as the original embeddings. Obtain normalized sets of image embeddings $F'_I$ and text embeddings $F'_T$, then calculate the similarity matrix of the two using the Eq.20 for the final result prediction.

$$F'_I = \{I'_1, \dots, I'_i\} = \{IDWT(\mathcal{A}'_1), \dots, IDWT(\mathcal{A}'_i)\} \quad (18)$$

$$F'_T = \{T'_1, \dots, T'_j\} = \{IDWT(\mathcal{B}'_1), \dots, IDWT(\mathcal{B}'_j)\} \quad (19)$$

$$M_{WaveDN} = L2(F'_T)^T \cdot L2(F'_I) \quad (20)$$

## 4 EXPERIMENTS

In the experimental section, we conducted experiments on 14 datasets with various VLM backbones for two downstream tasks. We aim to discuss and validate the following point through experimentation. 1) Can WaveDN be widely applied to multiple downstream tasks with different datasets and adapt to various VLM encoder

---

**Algorithm1: Hierarchical normalization**

**Input:** Images coefficients set A, texts coefficients set B, image average coefficients $\mathcal{A}_{Avg}$, text average coefficients $\mathcal{B}_{Avg}$, $\lambda$ constant

**Output:** Hierarchical normalized image coefficients set A′, hierarchical normalized text coefficients set B′.

**Procedure:** *HierarchicalNor*(Array X, Array Y, Array Z):
  *for* each index i from 1 to length of X do
    Initialize S as an empty array
    *for* each index j from 1 to length of Y do
    x = X[i][j]
    y = Y[j]
    //Calculate j-th wavelet coefficients' similarity.
    z = x · y
    k = x - z · y ·$\lambda$
    Append k to S
    end *for*
    Append S to Z
  end *for*
  Output Z
 End *Procedure.*
Initialize A′, B′ as empty arrays.
A′ = *HierarchicalNormalization*(A, $\mathcal{A}_{Avg}$, A′).
B′ = *HierarchicalNormalization*(B, $\mathcal{B}_{Avg}$, B′).
**Output:** A′, B′.

---

structures? 2) Compared to current advanced VLM testing enhancement methods, does our method have superiority? Does WaveDN mitigate the deficiencies of DN? 3) Will our method be affected by parameter settings? How much impact will the selection of sampling quantity and related wavelet parameters have on the effectiveness of WaveDN? Does the method possess robustness? 4) During the testing phase, is the computational cost significant for applying wavelet transform and hierarchical normalization to all embeddings? Does WaveDN result in a substantial computational overhead?

### 4.1 Downstream tasks

*4.1.1 Zero-shot classification.* This task involves using a pre-trained VLM model directly to predict the category to which the test samples belong. Our experiments include zero-shot recognition of both image and audio data. For image data, each category is represented by a fixed text template input to the text encoder to obtain the corresponding feature representation. The image encoder encodes the image to get the image feature representation, then calculates the similarity with the text embedding to determine the category with the highest similarity as the prediction result. The image dataset used includes: ImageNet1K[30], Cifar100[31], SUN397[33], Standford Cars[32], Flowers102[37], Food101[36], DTD[34], OxfordPets[35], EuroSAT[38] and FGVCAircraft[39]. For audio data, the only difference from images is the use of an audio encoder for feature extraction. The audio dataset used includes UrbanSound8K[41] and ESC-50[40]. To ensure the correctness of the test phase, we maintain a non-overlapping folds partition consistent with the one used

**Table 1: Zero-shot Top1 Accuracy on 10 Image Datasets**

| | ImageNet[30] | Cifar100[31] | Cars[32] | SUN397[33] | DTD[34] | Pets[35] | Foods[36] | Flowers[37] | EuroSAT[38] | Aircraft[39] | Average |
|---|---|---|---|---|---|---|---|---|---|---|---|
| CLIP(Vit-B32)[4] | 59.13 | 64.18 | 58.55 | 57.06 | 43.99 | **87.17** | 82.29 | **66.75** | 45.21 | 19.20 | 58.35 |
| CLIP+DN[14] | 58.61 | 65.20 | 58.99 | 58.24 | 45.30 | 86.77 | 83.11 | 65.80 | 49.02 | 19.92 | 59.10 |
| CLIP+WaveDN(ours) | **59.26** | **65.42** | **59.21** | **58.48** | **46.16** | 87.16 | **83.32** | 66.71 | **50.60** | **20.29** | **59.66** |
| CLIP(RN50)[4] | 55.84 | 40.70 | 56.21 | 58.49 | 41.97 | 85.54 | 77.16 | 65.75 | 37.51 | 16.98 | 53.62 |
| CLIP+DN[14] | 55.95 | 43.52 | 55.93 | **59.25** | **43.29** | 85.14 | 78.50 | 65.80 | 40.01 | 17.70 | 54.61 |
| CLIP+WaveDN(ours) | **56.08** | **45.99** | **56.33** | 59.00 | 43.14 | **86.07** | **79.18** | **66.06** | **45.04** | **17.90** | **55.48** |

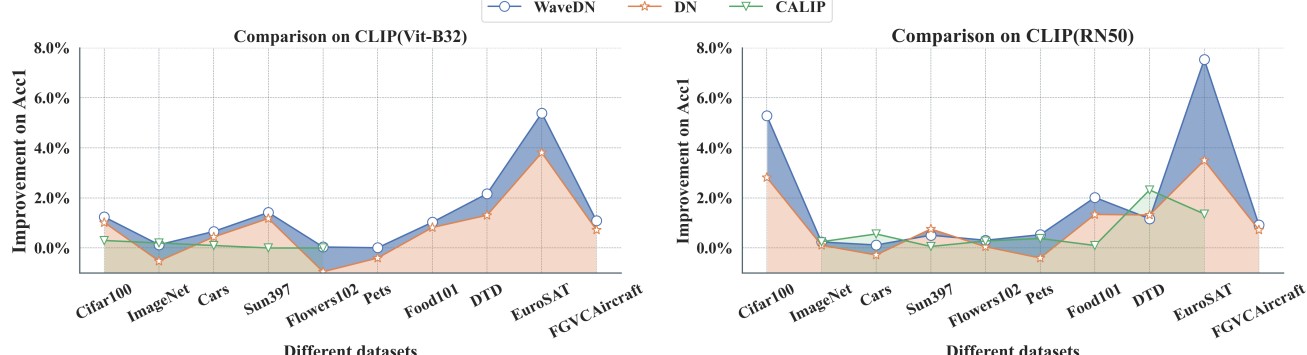

**Figure 4: Improvement in zero-shot top1 accuracy across 10 image datasets. WaveDN has demonstrated superior performance on two CLIP image encoders, outperforming or matching DN on all datasets.**

**Table 2: Zero-shot Top1 Accuracy on 2 Audio Datasets**

| | ESC-50[40] | UrbanSound8K[41] | Average |
|---|---|---|---|
| AudioCLIP(Full-training)[42] | 66.95 | 63.97 | 65.46 |
| AudioCLIP+DN[14] | 68.53 | 66.53 | 67.53 |
| AudioCLIP+WaveDN(ours) | **72.02** | **69.31** | **70.67** |
| AudioCLIP(Partial-training)[42] | 66.80 | 63.55 | 65.18 |
| AudioCLIP+DN[14] | 67.39 | 64.63 | 66.01 |
| AudioCLIP+WaveDN(ours) | **69.96** | **66.15** | **68.06** |

by the original authors for both audio datasets. For all the zero-shot experiments, we report Top1 accuracy(Acc@1) on each dataset.

*4.1.2 Cross-modal Retrieval.* Cross-modal querying task involves using a sample from one modality to query samples from another modality. It can be said that the classification task is a subtask of the cross-modal querying task. In our experiments, we conducted three cross-modal zero-shot querying experiments: text-to-image, image-to-text, and text-to-audio. For image-related tasks, we utilized the Flicker30k[43] and the COCO[44], while for audio-related tasks, we used the UrbanSound8K[41] and ESC-50[40] datasets. For retrieval experiments, the setup details for image-related datasets align with DN[14], while for audio-related datasets, our setup aligns with AudioCLIP[42]. We report the results of each experiment in terms of recall@1(R@1), recall@5(R@5) and recall@10(R@10).

## 4.2 Experimental Settings

During the experiment, we extensively tested various image and audio encoders. For image-related tasks, we utilized two different CLIP encoders, VIT-B32 and RN50, with implementation, image data preprocessing, and pretraining weights sourced from CLIP[4]'s open-source content. For audio-related tasks, we employed the model implementation, training weights, and audio data sampling and preprocessing methods publicly available in AudioCLIP[42], conducting experiments with both full-training and partial-training audio encoders. Across each dataset, in addition to testing with WaveDN, we also conducted experiments based on DN[14]'s open-source implementation and hyperparameter settings to test the DN on all datasets for deeply comparing the differences between the two methods.

Due to the need for both WaveDN and DN to sample test samples and then estimate distributions, and since random sampling results are inherently random, all our experimental results are obtained by conducting experiments on 5 randomly selected random seeds and averaging the results, aiming to minimize the randomness of the experimental results. During the distribution feature estimation, we sampled only 100 test samples for both, consistent with the approach used in DN[14]. For both methods, the samples used for distribution estimation are sourced from the dataset's test set to avoid introducing additional information and to simulate a realistic testing process, rather than from the validation set.

WaveDN uses Daubechies Wavelet as the wavelet basis for wavelet decomposition. Due to its hierarchical nature, it is well-suited for discrete wavelet decomposition. In the experiments, WaveDN consistently utilizes the "db6" wavelet basis with a decomposition level of 5. The impact of wavelet basis selection and decomposition levels on the effectiveness of WaveDN is extensively discussed in the

**Table 3: Retrieval Experiments on 2 Image Datasets**

| | MSCOCO (5K test set) | | | | | | Flicker30K (1K test set) | | | | | |
| | Image → Text | | | Text → Image | | | Image → Text | | | Text → Image | | |
| | R@1 | R@5 | R@10 | R@1 | R@5 | R@10 | R@1 | R@5 | R@10 | R@1 | R@5 | R@10 |
|---|---|---|---|---|---|---|---|---|---|---|---|---|
| CLIP(Vit-B32) [4] | 51.59 | 75.68 | 83.94 | 30.23 | 55.11 | 66.41 | 81.30 | 95.00 | 98.50 | 62.72 | 85.95 | 91.95 |
| CLIP+DN [14] | 51.86 | 75.90 | 83.85 | 33.38 | 58.53 | 69.36 | 83.38 | **96.41** | **98.54** | 65.91 | **88.20** | 93.27 |
| CLIP+WaveDN(ours) | **52.18** | **76.05** | **84.18** | **33.51** | **58.69** | **69.68** | **83.50** | 96.16 | 98.23 | **65.94** | 88.12 | **93.36** |
| CLIP(RN50) [4] | 50.72 | 75.08 | 83.36 | 28.54 | 52.70 | 64.10 | 82.20 | 95.59 | 97.89 | 59.88 | 85.91 | 91.86 |
| CLIP+DN [14] | 50.70 | 74.81 | 83.14 | **31.52** | 56.48 | 67.50 | **83.09** | 96.15 | **98.15** | 63.69 | **87.85** | **93.28** |
| CLIP+WaveDN(ours) | **50.87** | **75.35** | **83.66** | 31.42 | **56.54** | **67.61** | 82.28 | **96.25** | 97.77 | **63.71** | 87.24 | 92.95 |

ablation experiments section. The choice of wavelet basis and decomposition levels may have a slight impact on the experimental results. However, all combinations have a positive impact on the experimental results.

## 4.3 Experimental Results

*4.3.1 Zero-shot classification results.* The results of the zero-shot experiment for images are shown in Table. 1. We conducted a performance comparison on two types of CLIP image encoders, DN and WaveDN. WaveDN demonstrated significant improvements over the baseline across all datasets and outperformed DN. In the 14 datasets tested, WaveDN increased the performance by an average of 1.31% on Vit-B32 and 1.86% on RN50. The comparative effectiveness is illustrated in Figure. 4, where WaveDN shows superior overall performance, especially notable on the EuroSAT dataset with improvements of 5.39% and 7.53% on two different architectures. However, the results on Pets and Flowers indicate that the DN-related methods have negative optimization, possibly because CLIP (Vit-B32) itself has already adapted well to the data distribution of the test datasets. Additional operations on the already well-represented data may lead to information loss. Nonetheless, WaveDN reduces this negative impact compared to DN. Moreover, experimental results on a large number of datasets also show that this situation of negative optimization on the final results occurs less frequently.

The experimental results for zero-shot audio classification are shown in Table. 2. WaveDN demonstrates significant improvements on both Full-training and Partial-training audio encoders, with increases of 5.21% and 2.82% respectively compared to the baseline. Notably, on UrbanSound8k, the accuracy of full-training audio recognition increased significantly by 5.34%. WaveDN demonstrates good adaptability for audio-related classification problems.

*4.3.2 Cross-modal Retrieval results.* We conducted image-to-text and text-to-image retrieval experiments on MSCOCO and Flicker30k, with the experimental results shown in Table. 3. For all image encoder structures, using DN or WaveDN had a positive impact on all retrieval tasks. Moreover, WaveDN showed improvement over DN in most experimental results, reflecting WaveDN's support for various downstream tasks to some extent. We also observed that in some cases where the results were not ideal, such as the image-to-text retrieval of CLIP(RN50)+WaveDN on Flicker30k, DN performed better. This could be due to the unique data distribution

**Table 4: Retrieval Experiments on 2 Audio Datasets**

| | ESC-50 | | | US8k | | |
| | Text → Audio | | | Text → Audio | | |
| | R@1 | R@5 | R@10 | R@1 | R@5 | R@10 |
|---|---|---|---|---|---|---|
| AudioCLIP(Partial-training)[42] | 80.00 | **93.99** | 96.4 | 81.00 | 90.99 | 94.00 |
| AudioCLIP+DN [14] | 81.76 | 91.04 | 95.03 | 84.20 | 91.40 | 96.00 |
| AudioCLIP+WaveDN(ours) | **82.08** | 93.20 | **96.8** | **87.4** | **94.20** | **97.20** |
| AudioCLIP(Full-training)[42] | 82.00 | **95.19** | 96.40 | 84.00 | 92.00 | 96.00 |
| AudioCLIP+DN [14] | 81.36 | 92.80 | 95.51 | 88.40 | 96.60 | 97.80 |
| AudioCLIP+WaveDN(ours) | **83.11** | 94.80 | **96.80** | **89.00** | **97.00** | 97.80 |

of Flicker30k leading to insufficient feature estimation by WaveDN. However, in other retrieval experiments, WaveDN demonstrated good adaptability.

Experimental text-to-audio retrieval was conducted on ESC-50 and UrbanSound8k datasets, with the experimental results shown in Table. 4. Since both datasets only had a single text label for each audio, the audio-to-text retrieval experiment essentially overlapped with the zero-shot classification experiment. Therefore, we focused solely on text-to-audio retrieval. WaveDN demonstrated significant improvements across all results and managed to avoid the negative impact of DN methods on cross-modal retrieval. For instance, in the ESC-50 experiment with AudioCLIP (Partial-training), while DN improved the R@1 performance, there were decreases in R@5 and R@10. This was attributed to DN disrupting the intrinsic feature representation of samples. WaveDN, on the other hand, circumvented this issue and exhibited better performance across all metrics.

## 4.4 Ablation Study

*4.4.1 Convergence of Distribution Extraction.* Due to the need for WaveDN to sample multiple unlabeled test samples for estimating test data wavelet distributions, the quantity of sampled samples will have a significant impact on the distribution estimation results, thereby affecting the method's effectiveness. Therefore, we conducted sampling quantity experiments using CLIP(Vit-B32) to extract text and image features on Cifar100.

The experimental results, as shown in Figure. 5, indicate the similarity between the wavelet distribution estimated using the corresponding sample quantity and the true distribution. The true

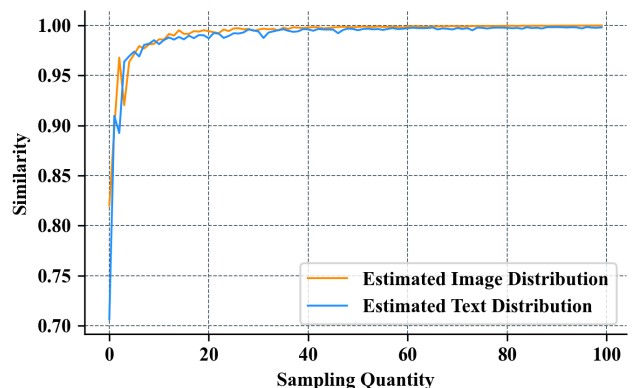

**Figure 5: Convergence of Distribution Extraction. Estimate wavelet distributions using different sample quantity. For all modal data, a stable and convergent distribution estimate can be obtained with only a small amount of unlabeled test data.**

distribution is obtained by performing wavelet transform and feature extraction on all samples. The results show that when the sampled samples approach 10, the similarity between the sampled estimated distribution and the true distribution reaches 97.5%, and when the sample quantity exceeds 40, the estimated distribution has already converged to the true distribution. These results indicate that only a small amount of unlabeled samples is needed to achieve effective distribution feature estimation. In our experiment, we uniformly used a sample quantity of 100 to ensure consistency with DN's sampling quantity.

**Table 5: Impact of Different Wavelet Bases and Decomposition Levels on CLIP(Vit-B32)'s Cifar100 Top1 Accuracy**

| db base/level | 1 | 2 | 3 | 4 | 5 | 6 | 7 |
|---|---|---|---|---|---|---|---|
| db2 | 65.24% | 65.29% | 65.35% | 65.27% | 65.35% | 65.29% | 65.27% |
| db4 | 65.32% | 65.34% | 65.27% | 65.31% | 65.29% | 65.21% | \ |
| db6 | 65.35% | 65.27% | 65.28% | 65.38 % | **65.42**% | \ | \ |
| db8 | 65.30% | 65.31% | 65.37% | 65.24% | 65.41% | \ | \ |
| db10 | 65.30% | 65.25% | 65.26% | 65.26% | \ | \ | \ |
| db12 | 65.27% | 65.31% | 65.25% | 65.27% | \ | \ | \ |
| db14 | 65.30% | 65.35% | 65.24% | 65.28% | \ | \ | \ |
| db16 | 65.25% | 65.28% | 65.17% | 65.28% | \ | \ | \ |
| db18 | 65.25% | 65.34% | 65.29% | \ | \ | \ | \ |
| db20 | 65.27% | 65.35% | 65.37% | \ | \ | \ | \ |

*4.4.2 The influence of wavelet bases and decomposition levels.* DWT and IDWT need to select the wavelet base used in the transformation and the number of decomposition layers. Lower decomposition levels are suitable for analyzing high-frequency detailed information, while higher decomposition levels are suitable for analyzing low-frequency overall characteristics, and different wavelet bases pay different attention to signal scales. To ensure that no extra information is lost in the decomposition process, the maximum decomposition levels of wavelet bases with different lengths are different.

The experimental results in Table. 5 show the influence of different selection schemes on the experimental results. All the results are averaged by experiments on 10 random seeds. Compared with the 64.1% top1 accuracy of CLIP(Vit-B32) on Cifar100, all the selection schemes of wavelet bases and layers after using WaveDN can improve the recognition accuracy, which proves the universality of our method. Different combinations of wavelet bases have a weak influence on the results, and we finally completed all our experiments with the combination of "db6" and "level 5", which is a combination with good adaptability to the characteristics of embedding.

*4.4.3 Computational complexity experiment.* During the testing phase, WaveDN incurred additional computational costs by performing extra operations on all samples. The main computations involved wavelet transformation, hierarchical normalization, wavelet inverse transformation, convolution operations, and additional similarity calculations for each sample. However, the experimental results in Table. 6 indicates that the additional computational cost introduced by WaveDN during testing is almost negligible compared to the costs of feature extraction and similarity calculation. When performing zero-shot image classification on the Cifar100 dataset, the computational overhead introduced by WaveDN averages only 0.0001 GFlops per sample. Compared to the original calculation method without WaveDN, the computational overhead for CLIP (Vit-B32) increased by only 0.0022%, and for CLIP (RN50) it increased by just 0.0017%. This further demonstrates the efficiency of our proposed method, as it only requires a small amount of computational overhead to enhance the performance of VLM on multiple downstream tasks.

**Table 6: Computation Overhead on Cifar100 Test Set**

| Method | Total Operations(GFlops) | Per Image Operations(GFolps) |
|---|---|---|
| CLIP(Vit-B32) | 44,649.95 | 4.4650 |
| CLIP+WaveDN | 44650.94 | 4.4651 |
| CLIP(RN50) | 58830.58 | 5.8831 |
| CLIP+WaveDN | 58831.57 | 5.8832 |

## 5 CONCLUSION

In this paper, we introduce WaveDN, a parameter-free, training-free method for enhancing the zero-shot capability of VLMs. WaveDN innovatively interprets the embeddings as a time series signal, subsequently applying discrete wavelet transform for feature extraction. By estimating feature distribution and employing hierarchical normalization, WaveDN aligns with the InfoNCE loss during the testing phase, without compromising the original sample feature representation. WaveDN requires only a small number of unlabeled test samples, with computational costs negligible compared to the entire inference phase. Experiments and results in multimodal settings also demonstrate WaveDN's beneficial effects on multiple downstream tasks with various VLMs. WaveDN is influenced by sample distribution in specific scenarios, indicating that future research could beneficially focus on this aspect. Improved methods for sampling and distribution estimation may exist and warrant exploration.

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
