# OpenReview forum: "WaveDN: A Wavelet-based Training-free Zero-shot Enhancement for Vision-Language Models"
_acmmm.org/ACMMM/2024/Conference — MM2024 Poster_

### Official Review · Reviewer_ufbn · 2024-05-21

**Rating:** 2
**Confidence:** 3

**Summary:**

This paper proposes a plug-and-play module between the CLIP model and the final similarity metric. It applies the wavelet transform to the clip embeddings and then adopts a hierarchical normalization. The idea follows previous DN work in that it only changes the metric calculation so it just needs to sample from the test dataset instead of fine-tuning and retraining the CLIP model. The work conducted plenty of experiments and achieved somehow potential results.

**Strengths:**

1. The paper is well presented.
2. The formulation of the method is formal and complete, with enough ablation studies to verify the components.
3. Lots of experiments demonstrate their efficacy (but not significant).

**Limitations:**

Though the paper is complete and the experiments are fair enough. I still have several concerns.

1. The novelty of this work sounds quite incremental to me, compared to the previous work [1]. Basically, this work just substitutes the mean with a wavelet distribution from a small portion of the test samples. They normalize the text and image distribution in the wavelet domain rather than direct multiplication in [1].
2. Further, the slight enhancement on many datasets strengthens my concern regarding the incremental idea. In Tab1, their improvements are less than 2%, mostly less than 1.5%. Further for large-scale datasets in Tab3, their effects compromise compared to DN.
3. Introducing wavelet transform as well as the necessary inverse transform will bring additional computation costs, I do not see any ablations regarding such computation costs analysis. Please add more ablations here.
4. There is a presentation deficiency on L128 Page 2. I can not see any proof that the degeneration contributed to the representation quality. What I can see from Fig2 is that after using your method, it does somehow enhance the accuracy for some categories. But nothing to do with representation. So I am concerned about your design out of this motivation that is to use wavelet to enforce representation ability. And this motivation is further diminished by the slight performance.
5. Some of the arguments are not convincible. 1) In the abstract, why the training of an additional parameter module would impair the existing transferability of VLMs? From my perspective, those adapters are quite flexible to downstream tasks. 2) On L120 of Page 2, how to prove the additional information loss, any proof about these, citations and ablations and experiments. and what are the compromised features, more explanation is required.
6. Related works should be enhanced. 1) the distinction between your work and previous works is not clear, further, you should better present your novelty. 2) Further, which wavelet method do you choose, and why do they have more semantic modeling capability? What is the purpose you choosing it? Compared to others, why is it the one? More explanation here.

[1]Test-Time Distribution Normalization for Contrastively Learned Vision-language Models

**Suitability:**

2

---

### Official Review · Reviewer_cptt · 2024-05-22

**Rating:** 4
**Confidence:** 4

**Summary:**

This paper introduces a wavelet-based distribution normalization method for adapting current VLMs, which is a training-free approach. The method is based on previous work DN and further takes wavelet transform to minimize information loss during the feature normalization. Experiments on two tasks, namely zero-shot classification and cross-modal retrieval demonstrate promising results of the proposed method.

**Strengths:**

1.	The paper is well-organized and easy to follow.
2.	The motivation to improving DN is reasonable and clearly presented.
3.	The idea of incorporating DWT into feature normalization is interesting.

**Limitations:**

Based on the statements in the paper, my main concern is that the experiments did not thoroughly verify the author's motivation.
1.	The motivation of using DWT is to minimize the information loss in feature normalization. Why and how can DWT achieve this purpose? Please elaborate on this.
2.	Based on 1., Could you provide visualization analysis to validate whether normalizing after DWT can reduce information loss compared to DN?
Some additional concerns are about the performance:
3.	Please make a performance comparison to CALIP.
4.	In Table 1, the performance increase with ViT-B/32 backbone compared to DN is limited on most datasets (less than 1%).
5.	I am curious about the performance comparison between your method and the prompt tuning approach (e.g., CoOp). If possible, could you provide a simple comparison?
Based on the above strengths and concerns, my initial rating is borderline accept.

**Suitability:**

2

---

### Official Review · Reviewer_S9tS · 2024-05-24

**Rating:** 5
**Confidence:** 3

**Summary:**

This paper proposed a method, WaveDN, to enhance the zero-shot transfer capability of Visual Language Model (CLIP) using the wavelet transform. WaveDN achieves the alignment of InfoNCE loss and test data feature distribution by hierarchically normalizing the mean wavelet distribution of a few test data samples directly in the test phase, without introducing new model parameters and without training. Experiments on 14 image and audio datasets demonstrate the effectiveness and superiority of the approach.

**Strengths:**

1. WaveDN applies wavelet transform before the similarity calculation process of VLM (CLIP), which can realize more accurate distribution feature extraction compared to DN, and has a certain novelty.
2. WaveDN applies wavelet-based feature extraction, which maximizes the preservation of local information about each sample, and hierarchical normalization, which, by controlling the strength of each level of normalization, introduces distributional information while better preserving more specific information about the test sample.
3. WaveDN is used directly in the testing phase without adding new model parameters or training, which does not affect the zero-shot capability of the VLM (CLIP) itself, and the computational overhead of WaveDN is very low.

**Limitations:**

1. In line 491“Also, there is a fixed constant 𝜆 that will be used in the normalization process, whose value is determined through experiments.”, during the hierarchical normalization stage of WaveDN, please specify the method of obtaining the constant λ. Will WaveDN be sensitive to this constant λ? Is it possible to do a set of experiments to demonstrate the effect of different λ on the retention effect of distribution information and test sample-specific information?
2. While Table 1 in the main text demonstrates that WaveDN has a performance improvement in the DTD texture dataset compared to DN and Baseline, in Figure 1 in the Appendix, from a category perspective, WaveDN shows a large additional negative impact in the -0.06~0.00 interval compared to DN in the DTD texture dataset (as shown in the red box) and a large additional positive impact in the 0.12~0.18 interval (as shown in the green box), does this suggest that in some cases WaveDN will have a more extreme impact on some of the categories tested? Is this related to the randomness of the sampling?

**Suitability:**

3

---

### Official Review · Reviewer_SWDY · 2024-05-25

**Rating:** 4
**Confidence:** 3

**Summary:**

This paper proposes a parameter-free, training-free, and computationally efficient VLM zero-shot enhancement method. It achieves similarity calibration by extracting different levels of wavelet coefficients from an unlabeled test set and performing hierarchical normalization on the test set data features.

**Strengths:**

Advantages:
1.	The method is straightforward and achieves relatively good results by normalizing the distribution of the dataset from the perspective of wavelet coefficients.
2.	The experimental content is relatively complete, with comparative experiments conducted on 14 datasets, and ablation studies on various components of the method are also quite comprehensive and thorough.

**Limitations:**

Disadvantages:
1.	There are concerns regarding the experimental setup. The number of comparative methods is too few; the paper should compare its method with other approaches mentioned in related work, such as those in references [13], [17], [18], and [19].
2.	Generally, zero-shot learning allows access to only one test sample at a time, whereas this paper’s setup allows access to most of the test samples at once.

**Suitability:**

3

---

### Meta-Review · Area_Chair_ogNf · 2024-06-30

**Recommendation:** Accept (Poster)
**Confidence:** 4

**Metareview:**

This paper introduces WaveDN, a method to enhance the zero-shot transfer capability of the Visual Language Model (CLIP) using wavelet transform. WaveDN aligns the InfoNCE loss with the test data feature distribution by hierarchically normalizing the mean wavelet distribution of a few test samples directly during the test phase, without adding new model parameters or requiring additional training. Extensive experiments across various benchmarks demonstrate the effectiveness of the design. Although Reviewer ufbn questioned the novelty of this work, the authors have addressed most of the concerns during the rebuttal. Besides, most reviewers acknowledge the motivation, presentation, and extensive validation of the proposed method. Consequently, the AC recommends to accept this submission.